# Algae Bloom and Decomposition Changes the Phosphorus Cycle Pattern in Taihu Lake

**Chaonan Han *** , **Yan Dai, Ningning Sun, Hao Wu, Yu Tang and Tianhao Dai**

School of Civil Engineering, Nanjing Forestry University, Nanjing 210037, China
* Correspondence: hcn_125@163.com; Tel.: +86-13022593296

**Abstract:** Algae bloom event, an extreme ecological imbalance that the water environment experiences, changes the phosphorus (P) cycle in the aquatic environment, which makes the lake maintain a long-term eutrophication and frequent algae bloom state. This study compared P form characteristics and bacteria community structures in the aquatic environment of the cyanobacteria area and non-cyanobacteria area of Taihu Lake, aiming to clear the new P cycle pattern disturbed by algae bloom and decomposition processes. Compared with P forms in mediums of the middle of the lake and the east of the lake, there were higher concentration levels of total particulate P (TPP) in water, organic P (OP) in suspended particles, iron bound P (FeP) in sediments and phosphate ($PO_4^{3-}$) in the pore water of Meiliang Bay, the cyanobacteria area. OP form was the dominant P fraction in suspended particles that occupied 69% in particulate total P, but OP proportion in sediments decreased to 26% of sediment total P, which indicated the strong occurrence of OP mineralization in sediments. The higher concentration and proportion of FeP in sediments of Meiliang Bay suggested the intensified effects of algae bloom and decomposition on sediment FeP accumulation. In Meiliang Bay, the positive correlation between $Fe^{2+}$ and $PO_4^{3-}$ in pore water and the higher diffusion fluxes of $Fe^{2+}$, $PO_4^{3-}$ from pore water to overlying water (0.45, 0.65 mg/m²·d) than that in the other lake areas also suggested the intensified effects of algae bloom and decomposition on FeP reductive dissolution in sediments accompanying sediment P remobilization. Moreover, there were higher concentrations of labile sulfide and high relative abundances of iron reducing bacteria (FRBs), sulfate reducing bacteria (SRBs) in sediments of Meiliang Bay. Results suggested that algae bloom event changed the natural P cycle in aquatic environment through intensifying the pathways of sediment OP mineralization, FeP accumulation and FeP reductive dissolution, which were mainly driven by the coupled factors of anoxic sediment condition, SRBs and FRBs activities. In addition, $PO_4^{3-}$ diffusion from pore water to overlying water in the east of the lake may be prevented for its much higher Fe/P ratio (8.06) and $Fe^{2+}$ concentrations in pore water, which may form a P-adsorbing barrier of iron oxides in the interface between pore water and overlying water. This study enhances the understanding of the vicious P cycle pattern in the aquatic environment driven by algae bloom and decomposition, which should be considered when conducting eutrophication prevention and control measures on lakes.

**Keywords:** phosphorus cycle; algae bloom; algae decomposition; Taihu Lake; aquatic environment

## 1. Introduction

Phosphorus (P) is an essential nutrient for living organisms, which plays a key role in regulating water quality, primary production and biogeochemical cycle in aquatic environment [1,2]. Excessive P inputs can easily trigger water eutrophication and algae blooms in lakes or reservoirs, such as Taihu Lake and Erie Lake [3,4]. If a water body has already been in an eutrophic state, reducing external P loads can not always lead to rapid recovery from lake eutrophication and algae blooms [5]. Some viewpoints showed that internal P release from sediments resulted in the delayed response [6,7].

Iron (Fe) plays an important role in controlling the mobility of P in sediments [8,9]. Soluble reactive P tends to be chemically or physically adsorbed on ferric iron oxides/hydroxides

in sediments, whereas it could also be released due to iron oxides/hydroxides reductive dissolution [10]. In sediments, decomposition of organic matter driven by micro-organisms exhausts oxygen resulting in anoxia condition, in which iron oxides/hydroxides reduction may occur easily. Except of chemical factors, micro-organisms' activities can also drive the reduction of iron oxides/hydroxides. Iron reducing bacteria (FRB), such as *Geobacter*, *Geothrix* and *Shewanella* in freshwater sediments, works for ferric iron oxides reduction to obtain energy [11], accompanying with the dissolution of phosphate ($PO_4^{3-}$) and ferrous iron ($Fe^{2+}$) from sediments. In contrast, iron oxidizing bacteria (FOB) acts on the formation of iron oxides/hydroxides [12], limiting sediment P mobility.

Sulfate reduction in sediments driven by sulfate reducing bacteria (SRB) may be coupled with iron oxides/hydroxides reductive dissolution [13]. High abundance of SRB was simultaneously observed with the high release of $Fe^{2+}$ and $PO_4^{3-}$ in sediments of the cyanobacteria-dominated zones [14]. Sulfide ($S^{2-}$), the production of sulfate reduction, reacted with $Fe^{2+}$ to form the precipitation of iron sulfide (FeS), which may promote iron oxides/hydroxides reduction in sediments. In addition, organic matter in sediments can also influence Fe, S and P biogeochemical cycles in sediments. Xiao's study observed that soil organic carbon was positively correlated with $Fe^{2+}$ and $S^{2-}$, owing to the crucial role of soil organic carbon favoring iron reduction and sulfate reduction [15].

Algae cell overgrowth produces algae bloom phenomenon in freshwater lakes, which is an important intermediary in the biogeochemical cycles of carbon, nitrogen and P [16–18]. In algae growth period, amounts of soluble P in water are first absorbed into algae cells, and then transported to sediments with algae death and deposition, resulting in sediment P accumulation [19]. Then, algae decomposition in sediments affects P remobilization and transformation. Chen's study indicated that algae decomposition drove iron oxides/hydroxides reduction accompanying with the dissolution of $PO_4^{3-}$ from sediments, and algae cell lysis also released some organic P into pore water [9]. Several studies observed the correlation among P, Fe and S remobilization in sediments [8,20]. Therefore, knowledge about the transport and transformation of P among different water environment mediums is needed to better understand ecological effects of algae blooms in aquatic environment.

Taihu Lake is the third largest freshwater lake in China and is a shallow water lake with an average water depth of approximately 2 m. In recent decades, the water body of Taihu Lake has suffered from eutrophication and serious algae bloom, posing a great threat to the water quality of local urban water supply source [4]. From 2007 to 2015 external P loads to Taihu Lake were reduced by nearly half, resulting in a slight decrease of P concentration in water [21], whereas algae blooms in Taihu Lake have still occurred every year since [22]. Why are water eutrophication and algae blooms in Taihu Lake so difficult to eliminate? The effects of algae bloom and decomposition process on P cycle in aquatic environment can answer this question, but it remains poorly understood.

This study hypothesized that algae bloom and decomposition processes changed the natural P cycle pattern in the aquatic environment through intensifying the pathways of organic P (OP) formation-mineralization and iron bound P (FeP) accumulation-dissolution in sediments. To test this hypothesis, the cyanobacteria area and the non-cyanobacteria area in Taihu Lake were selected to contrastively study their P cycle patterns. The major objectives were to (1) characterize P forms in water, suspended particles, sediments and pore water of these typical areas; (2) illustrate the changed P cycle pattern under the influence of algae bloom and decomposition processes; (3) discuss chemical-driven and biological-driven sediment P remobilizations. This study can enhance the understanding of the vicious P cycle pattern in aquatic environment driven by algae bloom and decomposition and thus explain the reason for the poor treatment effect of water eutrophication and algae bloom through reducing external P inputs of lakes in the long term.

## 2. Materials and Methods

### 2.1. Study Area

Three sample sites were selected in Taihu Lake, including Meiliang Bay, the middle of the lake and the east of the lake (Figure 1). Meiliang Bay (31°25.96′, 120°11.52′) is located in the north of Taihu Lake, whose water body has been eutrophic and frequently appeared cyanobacteria blooms. The middle of the lake (31°20.99′, 120°11.34′) is in the open water zone, occasionally disturbed by cyanobacteria blooms. The east of the lake is located in the east of Taihu Lake (31°3.49′, 120°27.12′), where cyanobacteria blooms rarely occur [23]. Samples were collected in August 2020.

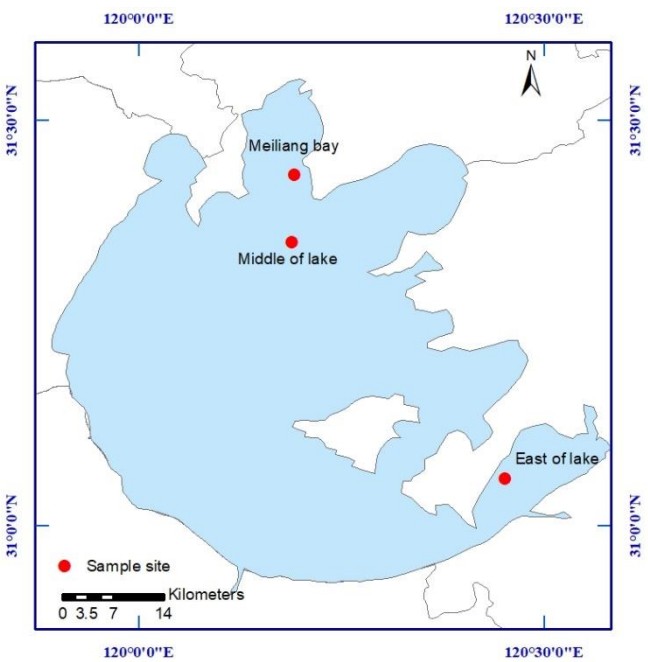

**Figure 1.** Location of sampling sites in Taihu Lake.

### 2.2. Sampling and Measurement Methods

Basic water quality parameters, including water temperature, dissolved oxygen (DO), pH, electric conductivity (EC), were measured by in situ sensors (Alalis PD320). Water samples were collected at depths of 0.5, 1 and 2 m below the water surface. Parts of water samples were filtered in the laboratory through glass fiber filters with a pore-size of 0.7 μm, and the filter membranes were extracted with 90% acetone solution to determine chlorophyll-a (Chla) by spectrophotometry method [24]. In addition, parts of water samples were filtered through acetate fiber filters with a pore-size of 0.45 μm, and the filtrate samples were stored at 4 °C for the measurement of orthophosphate ($PO_4^{3-}$) and total dissolved P (TDP). The original water samples were used to determine total P (TP). TP and TDP in water were detected using potassium persulfate digestion coupled with molybdenum blue [24], and $PO_4^{3-}$ in water were directly detected using molybdenum blue [24]. The difference between TP and TDP concentration in water is the total particulate P (TPP) concentration. The trapped particles by 0.45 μm filters were air-dried, mixed and sifted, respectively, for the measurement of inorganic P (IP) and organic P (OP) using the standardized SMT method [25].

Nine replicate sediment cores were sampled from three sample sites by gravity corer (11 cm × 50 cm, EasySensor Co., Nanjing, China). They were transported into laboratory within 6 h and maintained at 25 °C for employing HR-Peeper probes and DGT probes to determine $PO_4^{3-}$, ferric iron ($Fe^{2+}$) in pore water and labile sulfide ($DGT-S^{2-}$) in sediments. In addition, surface sediments were collected from three sampling sites and parts of them were kept frozen at −80 °C for the analysis of sediment bacterial community. The other surface sediments were air-dried and sifted through screen mesh with a pore-size of

0.15 mm for the measurement of IP, OP, iron bound P (FeP), calcium bound P (CaP) and sediment total P (S-TP) using the standardized SMT method [25].

### 2.3. Deployment of the HR-Peeper and DGT

The HR-Peeper probes and DGT probes were provided by Easysensor Co. in China (www.easysensor.net, accessed on 30 July 2020) and the preparation procedure was according to previous studies [26,27]. The HR-Peeper probe consists of 30 chambers, each with a volume of 200 μL. Before deploying into sediment cores, each chamber was filled with deionized water, and a 0.45 μm cellulose nitrate membrane entirely covered on the probe with 30 chambers. The HR-Peeper probe was deoxygenated overnight with nitrogen flushing. Then, it was inserted into the sediment core for 48 h to obtain the concentration equilibrium of dissolved substance between pore water and chamber water. After that, approximately 200 μL pore water sample was immediately sampled from each chamber, which was used to detect $PO_4^{3-}$ and $Fe^{2+}$ in pore water at a vertical resolution of 4.0 mm. The concentrations of $PO_4^{3-}$ and $Fe^{2+}$ in pore water were detected using molybdenum blue and phenanthroline colorimetric method [26].

The DGT probe with AgI binding gel was used to obtain DGT-$S^{2-}$ concentration in the sediment profile at a vertical resolution of 4.0 mm. DGT-$S^{2-}$ refers to the labile sulfide in sediments adsorbed by AgI binding gel in 24 h. The DGT probe was deoxygenated overnight with nitrogen flushing and then kept in oxygen-free water prior to deployment in sediment core. At 24 h after inserting the HR-Peeper probe into sediment core, the DGT probe was inserted back to back with the HR-Peeper probe into the sediment core. The DGT probe was kept in sediment core for 24 h and then was retrieved and rinsed with deionized water. The labile $S^{2-}$ adsorbed in AgI binding gel was measured by computer imaging densitometry technique [28] and then the DGT-$S^{2-}$ concentration in sediment core was calculated by using the approach described by Ding and Wang [29,30].

According to Fick's diffusion law, the apparent diffusion flux ($F_d$) of $Fe^{2+}$ and $PO_4^{3-}$ across the interface between pore water and overlying water (P-O interface) was estimated using the following equations [23], where $\varphi$ was the sediment porosity [31], $D_0$ was the diffusion coefficient of target analyte in water ($10^{-6}$ cm$^2$/s) [32], $\theta$ was the sediment tortuosity calculated by $\varphi$, $C$ was the solute concentrations of $PO_4^{3-}$ and $Fe^{2+}$ measured by the HR-Peeper probe (mg/L), $x$ is the sediment depth (m), $\frac{\partial C}{\partial x}$ was the concentration gradient of target analyte in sediment core at depths of $-20\sim5$ mm (from pore water to overlying water).

$$F_d = -\frac{\varphi \cdot D_0}{\theta^2}\frac{\partial C}{\partial X} \tag{1}$$

$$\theta^2 = 1 - In\left(\varphi^2\right) \tag{2}$$

### 2.4. DNA Extraction and 16S rRNA Gene High Throughput Sequencing

DNA was extracted from 0.5 g of freeze-dried sediments using E.Z.N.A.® soil DNA spin kit (Omega Bio-tek, Norcross, GA, USA). Then 2~5 ng of extracted DNA was subjected to polymerase chain reaction amplification using primers 338F (5′-ACTCCTACGGGAGGCA GCAG-3′) and 806R (5′-GGACTACHVGGGTWTCTAAT-3′), aiming to obtain the target gene 16S rRNA in the V3~V4 region. The library preparation was carried out using NEXTFLEX Rapid DNA-Seq Kit.

The quality control of raw sequence data was carried out using Trimmomatic software. The screening chimeras of clean sequences and operational taxonomic unit (OTU) clustering at 97% similarity were performed using Uparse (vsesion 7.0.1090, http://drive5.com/uparse/, accessed on 22 September 2022). Using the RDP Classifier, the taxonomic data was assigned to each representative sequence against the SILVA database (Release 132 http://www.arb-silva.de, accessed on 22 September 2022) at 70% similarity [33]. The above analyses were performed on QIIME platform [34].

The diversity of bacterial community was investigated using the Mothur software, which was showed as indices of Shannon, Simpson, ACE, Chao, and Coverage. The heatmap of bacterial community was made through Origin software. Major functional genes involved in inorganic P-solubilization and organic P-mineralization [35] were predicted and analyzed by PICRUSt2 (https://github.com/picrust/picrust, accessed on 22 September 2022) and Kyoto encyclopedia of genes and genomes (KEGG, http://www.genome.jo/kegg/, accessed on 22 September 2022).

## 3. Result

### 3.1. Chla Content and Water Quality Parameter in Water of Taihu Lake

In August 2020, Chla contents in water of Taihu Lake were shown in Figure 2, as well as pH, DO, EC and water temperature shown in Table S1. During the sampling period cyanobacteria blooms were breaking out in Meiliang Bay, thus its Chla contents in water (83~372 μg/L) were much larger than that in the middle of the lake (45~64 μg/L) and the east of the lake (18~24 μg/L) (ANOVA, $p < 0.05$). Chla contents in the vertical direction of Meiliang Bay decreased rapidly with the increase of water depth, while they remained basically stable in the vertical direction of the east of the lake and the middle of the lake (Figure 2). Large numbers of algal cells were concentrated in the surface water to compete for sunlight, thus algae photosynthesis caused a little higher DO and pH values in surface water of Meiliang Bay than the other two lake areas (Table S1).

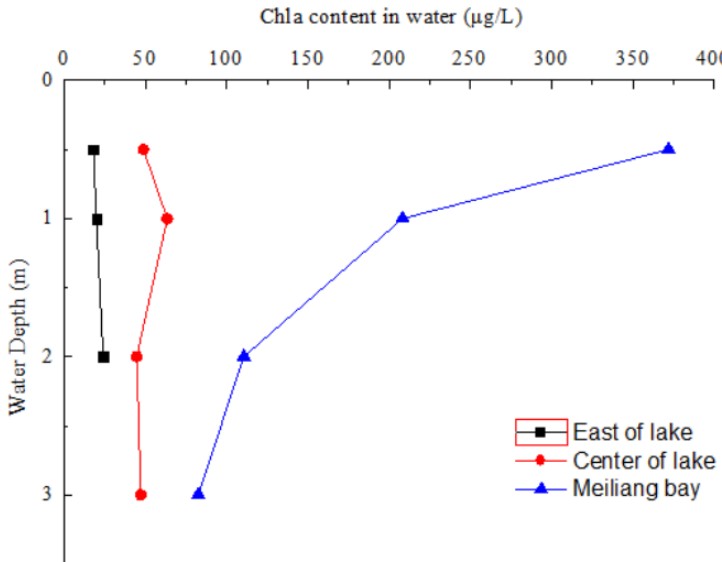

**Figure 2.** Chla contents in different water depths of Taihu Lake.

### 3.2. P Forms in Water, Suspended Particles and Sediments of Taihu Lake

In water, the determined P forms contained TP, TPP, $PO_4^{3-}$ and DOP (Figure 3). In water, TP concentrations were found in the order of 'Meiliang Bay (0.13~0.42 mg/L) > the east of the lake (0.18~0.28 mg/L) > the middle of the lake (0.08~0.12 mg/L)'. TPP was the dominant P form in water of three lake areas, which accounted for 72~96% of TP. Similar to the vertical distribution of Chla, concentrations of TP and TPP in water of Meiliang Bay also decreased gradually with the increase of water depth (Figure 3). The average concentrations of $PO_4^{3-}$ in water of Meiliang Bay, middle of the lake and east of the lake were 0.015, 0.007 and 0.005 mg/L, where the average DOP concentrations were 0.017, 0.012 and 0.005 mg/L, respectively.

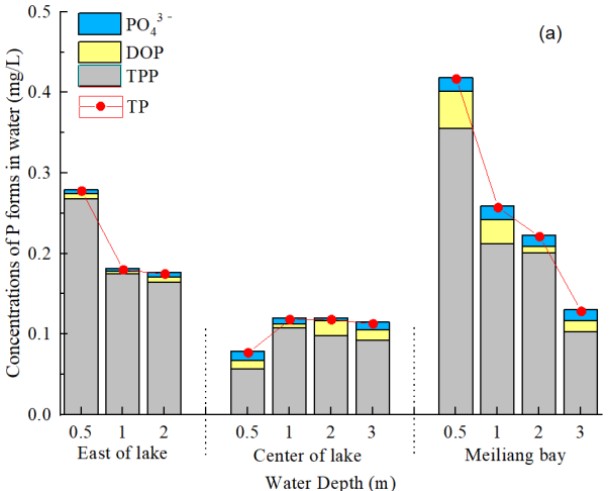
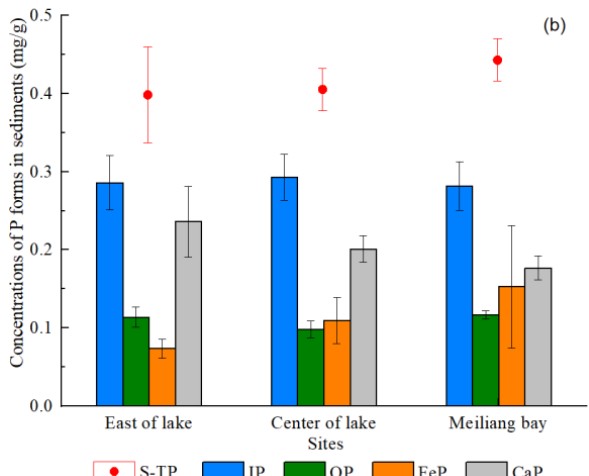

**Figure 3.** (**a**,**b**) Concentrations of P forms in water and sediments of Taihu Lake.

In suspended particles, concentrations of IP, OP forms in Meiliang Bay (3.21, 7.20 mg/g) were much higher than that in the Middle of the lake (0.55, 3.19 mg/g) and the east of the lake (0.04, 0.33 mg/g) (Figure S1). More than 69% of particulate P was in the form of OP in suspended particles of Taihu Lake.

In sediments, the determined P forms contained S-TP, IP and OP, FeP and CaP (Figure 3). 63~72% of S-TP was in the form of IP. In sediments, OP occupied only 24~28% in S-TP, much lower than OP proportion in suspended particles (over 69%). In addition, concentration of S-TP, FeP in sediments of Meiliang Bay (0.44, 0.15 mg/g) were relatively higher than that in the middle of the lake (0.41, 0.11 mg/g) and the east of the lake (0.40, 0.07 mg/g). CaP concentrations in sediments of Meiliang Bay were lower than that in the east of the lake and the middle of the lake. FeP and CaP accounted for 18~34% and 40~59% of S-TP in sediments.

### 3.3. High-Resolution Profiles of $PO_4^{3-}$, $Fe^{2+}$, DGT-$S^{2-}$ in the Interface between Pore Water and Overlying Water

$PO_4^{3-}$ concentrations in pore water followed the trend of being highest in Meiliang Bay, followed by the middle of the lake and the east of the lake (ANOVA, $p < 0.05$, Figure 4). DGT-$S^{2-}$ concentrations in sediments were also highest in Meiliang Bay, but $Fe^{2+}$ concentrations in pore water were highest in the east of the lake (Sig. $< 0.05$, Figure 4). In the vertical direction from the P-O interface to pore water, both $Fe^{2+}$ and $PO_4^{3-}$ concentrations in pore water of the three lake areas firstly raised and then declined (Figure 5). The highest values of $Fe^{2+}$ and $PO_4^{3-}$ were found at the depth of $-20$ mm in the east of the lake, $-10~-15$ mm in the middle of the lake, $-25~-30$ mm in Meiliang Bay. $Fe^{2+}$ concentrations were positively correlated with $PO_4^{3-}$ in pore water in the three lake areas (Sig. $< 0.05$).

The molar Fe/P ratios in pore water and $PO_4^{3-}$, $Fe^{2+}$ diffusion fluxes from pore water to overlying water of Taihu Lake were shown in Table S2. The average of molar Fe/P ratio in pore water of the east of the lake (8.06) was much higher than that in the middle of the lake (2.26) and Meiliang Bay (1.94). The $PO_4^{3-}$ diffusion flux from pore water to overlying water was the highest in Meiliang Bay (0.45 mg/m³·d), followed by the middle of the lake and the east of the lake. The $Fe^{2+}$ diffusion flux from pore water to overlying water in the east of the lake was also the highest among the three lake areas, but its $PO_4^{3-}$ fluxes was the lowest among the three lake areas (Table S2).

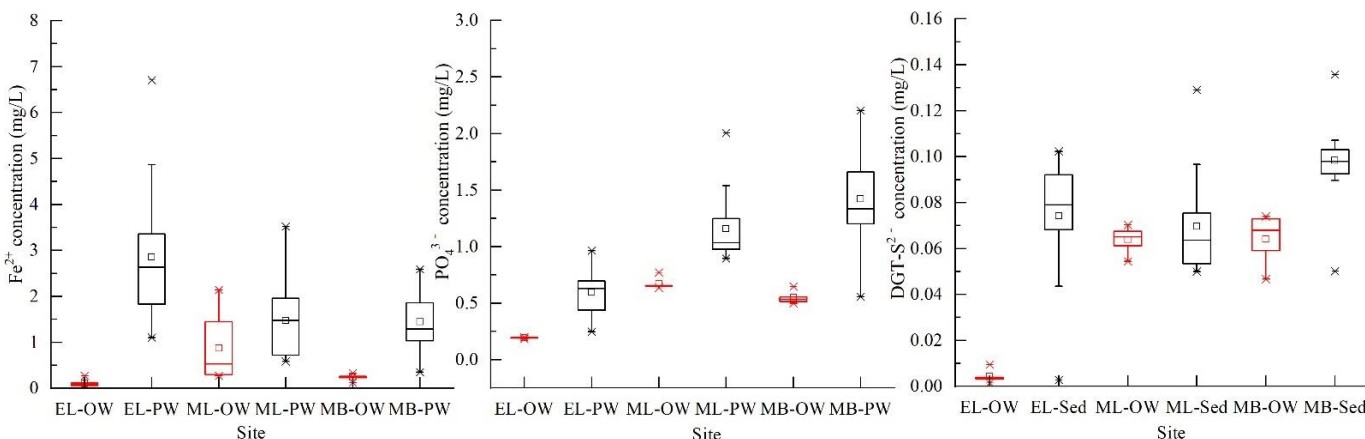

**Figure 4.** Concentrations of $PO_4^{3-}$, $Fe^{2+}$ in pore water and DGT-$S^{2-}$ in sediments of Taihu Lake. OW represents overling water, PW represents pore water, EL represents the east of the lake, ML represents the middle of the lake, MB represents Meiliang Bay.

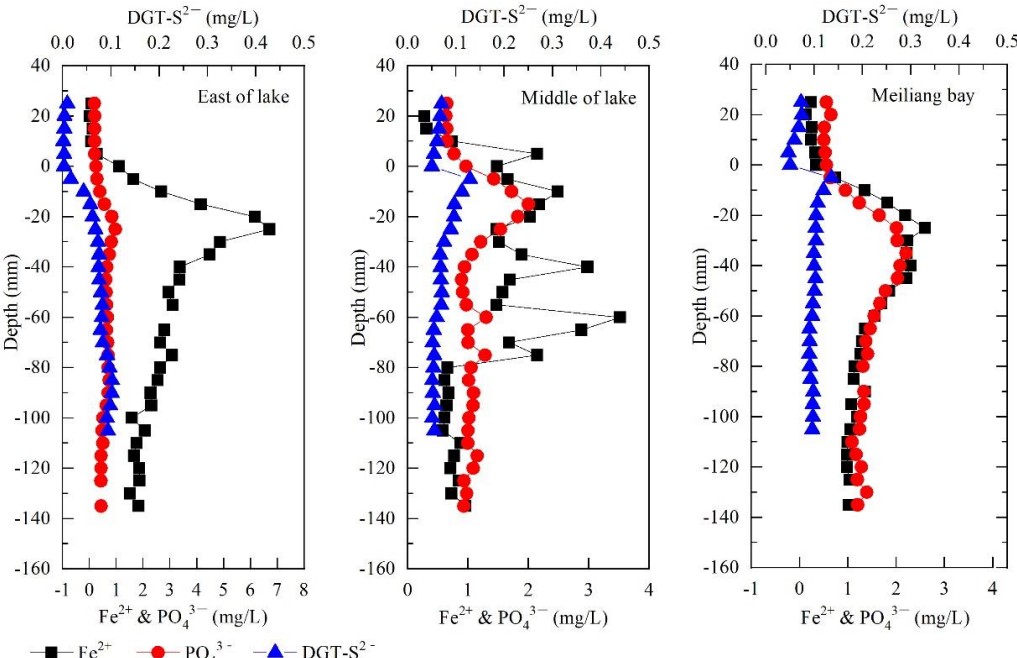

**Figure 5.** Concentrations of $PO_4^{3-}$, $Fe^{2+}$ in the profile of pore water and DGT-$S^{2-}$ in the profile of sediment of Taihu Lake.

### 3.4. Bacteria Community Structure of Taihu Lake

The identification of microbial communities from surface sediments of Taihu Lake by 16S rRNA obtained an average of 3892 OTUs. The sequence information calculated the coverage percentage, richness estimator (Chao and ACE), and diversity index (Simpson and Shannon) of the samples, which were listed in Table S3. On the whole, these parameters were no significant differences among the three lake areas, indicating that surface sediments in the three lake areas had similar richness and diversity of bacterial community in August 2020. The relative abundance of bacterial community was characterized at genus level, and the heatmap of top thirty bacterium was shown in Figure S2. The dominant bacterium at genus level in the three lake areas were *norank_c_Thermodesulfovibrionia* (5.23~6.61%), *norank_f_Anaerolineaceae* (3.04~8.53%), *norank_f_Steroidobacteraceae* (2.28~4.40%), *norank_o_ Vicinamibacterales* (2.11~3.94%), *norank_f_SC-I-84* (1.71~3.99%), etc.

In these top thirty bacterium, *norank_c_Thermodesulfovibrionia, norank_o_Syntrophobacterales*, Desulfobacca, norank_p_Desulfobacterota all belong to typical SRBs, indicating that the

sediment bacteria community of Taihu Lake had strong functions of sulfate reduction. In addition, some typical FRBs were also detected, such as *Anaeromyxobacter, Geothermobacter, Deferrisoma, Crenothrix*. Figure 6 showed the relative abundance of typical SRBs and FRBs in surface sediments of Taihu Lake. The relative abundances of typical SRBs and FRBs in the east of the lake (10.50% and 1.73%) were approximately equal to those in Meiliang Bay (10.31% and 1.19%), both of them were higher than those in the middle of the lake (7.70% and 0.52%).

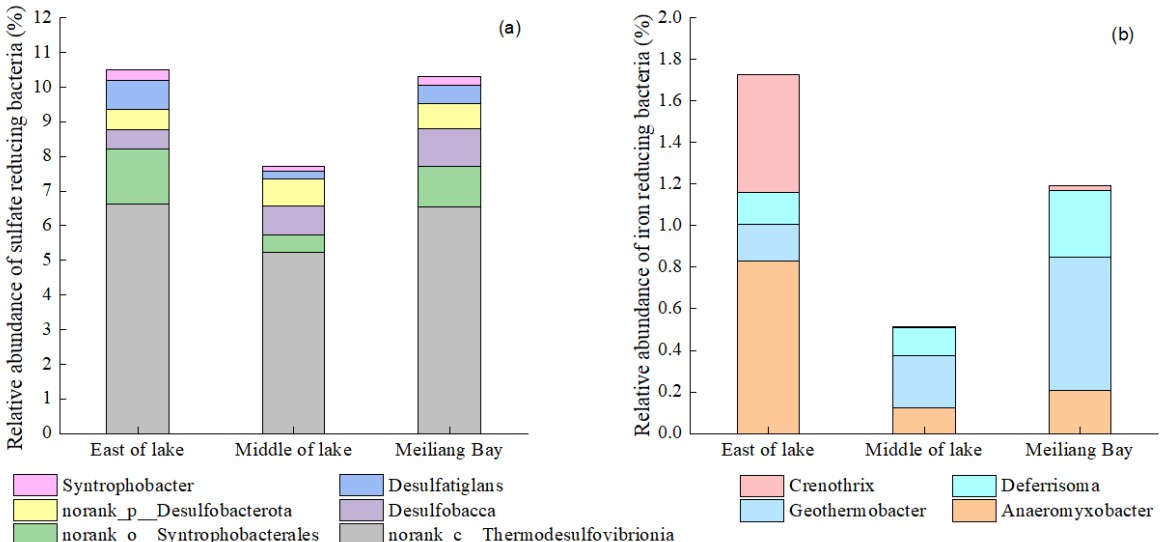

**Figure 6.** (**a**,**b**) Genus-level classification of typical sulfate reducing bacteria and iron reducing bacteria in surface sediments of Taihu Lake.

## 4. Discussion

### 4.1. Sediment OP Mineralization and FeP Accumulation Promoted by Algae Bloom and Decomposition

Based on P data in the three lake areas, concentrations of OP in suspended particles and sediment Fe-P in sediments of Meiliang Bay (the cyanobacteria area), were higher than those in the east of the lake and the middle of the lake (Figures S1 and 3). The spatial distribution of sediment P forms were similar to that in previous studies of Taihu Lake [36]. In general, sediment P loads in oceans, rivers, lakes source from exogenous P input and internal P accumulation [37]. The point sources throughout the Taihu Lake basin have been regulated over the past decades [21], which greatly reduced exogenous P loads that input in the Taihu Lake. Thus, the differences of sediment P forms in cyanobacteria area and non-cyanobacteria area should be largely caused by their internal P accumulation patterns.

Cyanobacteria is a kind of ancient bacterium with strong abilities to N fixation and P absorption [38,39]. In Meiliang Bay, the cyanobacteria area, relatively higher concentrations of $PO_4^{3-}$ in water guaranteed a sufficient quantity for cyanobacteria cell assimilation and growth [38,39]. These algae uptake process transported $PO_4^{3-}$ in water into algae cells, which can explain the extra high contents of IP and OP in suspended particles of Meiliang Bay (Figure S1). IP and OP in suspended particles of Meiliang Bay accounted for 31% and 69% of particulate P, but sediment IP and OP occupied 63% and 26% in S-TP, as shown in Figure S3. These suggested that particulate P forms were transformed in the migration process from suspended particles to sediments. These may be related to OP mineralization and FeP formation in sediments of Meiliang Bay.

When algae cells containing high OP content settle in sediments, cells decompositions driven by micro-organisms generate amounts of low molecular weight substances such as sugars, organic acids, proteins, organic nitrogen, OP, and so on [40,41]. Some OP produced from cell lysis such as phosphate monoester, phosphate diester, polyphosphates are easily transformed to dissolved inorganic P via enzymatic hydrolysis [42]. In surface sediments

of Meiliang Bay, the relative abundances of major functional genes involved in inorganic P-solubilization and organic P-mineralization [35] were a little higher than that in the other two areas (Figure 7), indicating sediment bacteria group had a relatively strong OP mineralization ability. Due to sediment OP mineralization driven by micro-organisms in Meiliang Bay, the proportion of OP in S-TP dropped sharply from 69% in suspended particles to 26% in sediments.

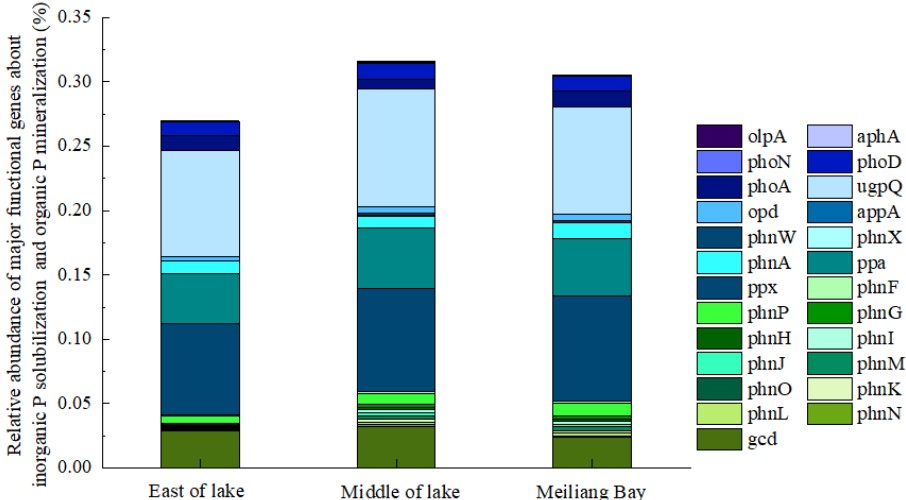

**Figure 7.** Major functional genes about inorganic P solubilization and organic P mineralization in surface sediments of Taihu Lake.

It was found that FeP concentrations in sediments of Meiliang Bay were relatively higher than that in the other lake areas (Figure 3b). We inferred that the high FeP in sediment of Meiliang Bay was mainly produced from amorphous Fe(III) (oxy) hydroxide adsorbing $PO_4^{3-}$ in pore water that was released from OP mineralization. Undoubtedly, the factor of algae bloom and decomposition played a key role in the accumulation process of FeP in sediments. Therefore, compared with the middle of the lake and the east of the lake (the non-cyanobacteria area), the relatively higher OP content in suspended particles, the higher FeP content in sediments of Meiliang Bay (the cyanobacteria area) reflected the net results between excess cyanobacteria cells deposition, extensive OP mineralization, sorption of dissolved P onto amorphous Fe(III) (oxy) hydroxide in the long time.

### 4.2. Sediment P Remobilization Driven by FeP Reductive Dissolution

Sediment P remobilization is largely influenced by Fe redox cycling through P adsorption by amorphous ferric iron oxides under aerobic condition as well as their reductive dissolution induced co-release of $Fe^{2+}$ and $PO_4^{3-}$ under anaerobic condition [43,44]. As a good electron acceptor in sediments, ferric iron oxides/hydroxides are more likely to accept electrons to occur chemical reduction [45]. For micro-organisms' activities, iron can act as both an electron donor such as $Fe^{2+}$ and as an electron acceptor such as $Fe^{3+}$ [46], the latter of which leads to biological reduction of ferric iron oxides/hydroxides. In Taihu Lake, the similar vertical distributions of $Fe^{2+}$ and $PO_4^{3-}$ in pore water and their positive correlations supported the occurrence of FeP reductive dissolution in sediments. However, concentration levels of $Fe^{2+}$, $PO_4^{3-}$ in pore water and their diffusion fluxes at the P-O interface were very different between the cyanobacteria area (Meiliang Bay) and the non-cyanobacteria area (the east of the lake).

Sediment P remobilization in Meiliang Bay seemed strongest among the three lake areas, showed by its highest concentrations of $PO_4^{3-}$ in pore water and $PO_4^{3-}$ diffusion flux from pore water to overlying water (Table S2). The synchronous vertical distribution of $PO_4^{3-}$ and $Fe^{2+}$ concentrations in pore water of Meiliang Bay confirmed the occurrence of FeP reductive dissolution. It can be seen from Figure 5 that the most active area of FeP

reductive dissolution in sediments of Meiliang Bay was concentrated at depths ranging from −10 mm to −60 mm. Thus, the extra accumulation of FeP in sediments of the cyanobacteria area just can be used as abundant materials for FeP reductive dissolution in sediments. However, the relative abundance of typical FRBs containing *Anaeromyxobacter*, *Geothermobacter*, *Deferrisoma*, *Crenothrix* [23] in sediments of Meiliang Bay was slightly lower than that in the east of the lake (Figure 6b), suggesting their limited ability of biological-driven ferric iron oxides/hydroxide reduction compared with that in the east of the lake.

Previous study had indicated that sulfate reduction driven by SRBs can also partly promote ferric iron oxides/hydroxide reduction, because some of ferric iron oxides/hydroxide may be mistakenly used as electron acceptor in the electron transport chain of sulfate reduction process [47]. The relative abundance of typical SRBs in sediment of Meiliang Bay was basically equal to that in the east of the lake, but the DGT-$S^{2-}$ contents in sediments of Meiliang Bay were higher than that in the east of the lake (Figure 5). The high DGT-$S^{2-}$ concentrations in sediments of Meiliang Bay suggested the occurrence of sulfate reduction and sulfide production, which occurred even prior to ferric iron oxides/hydroxide reduction [48,49]. In addition, based on the precipitation hydrolysis of $Fe^{2+}$ and $S^{2-}$, the formation of FeS precipitate can further promote ferric iron oxides/hydroxide reduction in sediments [50,51]. Therefore, both of anoxic sediment condition, SRBs and FRBs activities jointly acted on Fe(III) (oxy) hydroxide reduction accompanying sediment P remobilization of Meiliang Bay.

Strangely, pore water in the east of the lake had the highest $Fe^{2+}$ concentrations and the lowest $PO_4^{3-}$ concentrations among the three lake areas, and its average stoichiometric Fe/P ratio reached 8.06. For iron, we guessed that the sediments in our sampled area of the east of the lake were just richer in amorphous iron oxides, which may be not common in sediments of the east of the lake. The low P concentration levels in water, suspended particles, sediments and pore water of the east of the lake were common in the east of the lake (Figures 3 and 4), which were confirmed by other studies [36]. The low P concentrations in water of the east of the lake limited P uptake by algae cells and P adsorption by iron oxides in sediments. The high relative abundance of FRBs in sediments of the east of the lake meant their strong ability of Fe(III) (oxy) hydroxide reduction, explaining the extra $Fe^{2+}$ concentrations in pore water of the east of the lake. However, the limited accumulation of sediment FeP in the east of the lake suggested the limited amounts of FeP reductive dissolution, leading to the low $PO_4^{3-}$ concentrations in pore water of the east of the lake. In addition, based on the oxidative hydrolysis of Fe and the concomitant precipitation of P, the stoichiometric Fe/P ratio of 2 during the shift from anoxic to oxic conditions can be used as an indicator of P retention ability of lakes [52,53], but the average of molar Fe/P ration in pore water of the east of the lake was 8.06. Thus, the high $Fe^{2+}$ concentrations in pore water of the east of the lake may promote the formation of Fe(III) (oxy) hydroxide in the P-O interface, which could effectively reduce the diffusion flux of $PO_4^{3-}$ from pore water to overlying water (Table S2).

Compared with Meiliang bay and the east of the lake, Fe(III) (oxy) hydroxide reduction driven by micro-organisms seemed relatively weak in sediments of the middle of the lake for their low abundances of FRBs and SRBs (Figure 6). However, $PO_4^{3-}$ concentrations in pore water of the middle of the lake were still positively correlated to $Fe^{2+}$, and were higher than the east of the lake and lower than Meiliang Bay. It can be seen from Figure 5 that the most active area of FeP reductive dissolution in sediments of the middle of the lake was concentrated at depths ranging from −10 mm to −30 mm. These implied that chemical reduction of FeP under anoxic and reducing environment played a key role in sediments of the middle of the lake, by analogy, Meiliang Bay and the east of the lake as well.

### 4.3. Reason about Algae Bloom in Eutrophic Lakes Endless for a Long Time

In general, dissolved P in water is firstly transformed to particulate P by adsorbing onto particles or absorbing into plankton, and then particulate P settles in sediments and experiences morphological conversion including OP mineralization, FeP formation and

dissolution and so on, finally trending to form calcium phosphate minerals [54]. This natural pattern of P cycle in water environment remains stable over a long time, but the frequent occurrence of algae bloom slowly changed this pattern, creating a new P cycle pattern in lakes.

According to the above discussions, the P cycle patterns in the cyanobacteria area (Meiliang Bay) and in the non-cyanobacteria (the east of the lake) of Taihu Lake can be revealed (Figure 8). In the cyanobacteria area of Taihu Lake, algae bloom and decomposition processes intensified the cycle pathway of sediment FeP accumulation and FeP reductive dissolution coupled with sulfate reduction and FeS formation, forming a vicious P cycle among water, algae, sediment and pore water. The vicious P cycle in cyanobacteria area has been difficult to break naturally, so annual occurrence of algae bloom is hard to eliminate within a few decades. There were also similar development trends of the P cycle pattern in the middle of the lake, the open water zone with light cyanobacteria blooms. Whereas, maybe due to the high $Fe^{2+}$ contents and the extra Fe/P molar ratios in pore water of the east of the lake, transformation between ferric iron and ferrous iron may form a barrier in the P-O interface, preventing $PO_4^{3-}$ from spreading up overlying water.

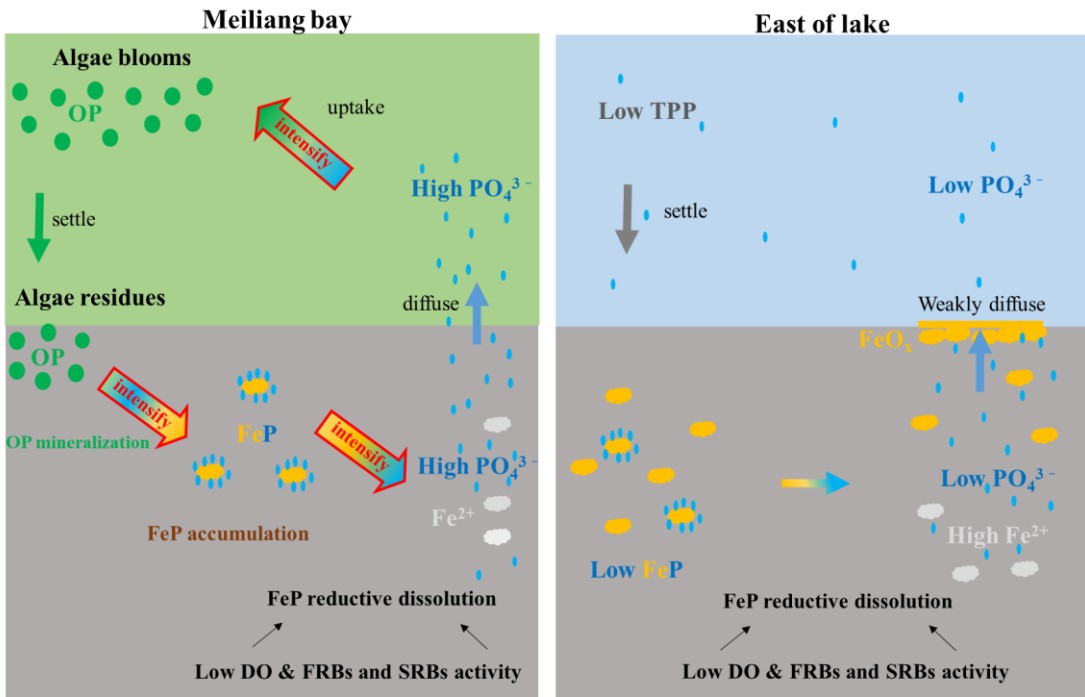

**Figure 8.** The P cycle patterns in Meiliang Bay and the east of the lake.

Given all this, Meiliang Bay in Taihu Lake is the chief consideration in breaking the vicious P cycle pattern, and thus eliminating algae blooms in the long term. Of course, controlling external P input of lake is effective on reducing P pollution in lakes in the long term, and it would take decades or even centuries to produce a new P cycle with low P concentration levels [21]. How to reduce P concentration level quickly and eliminate frequent algae blooms in lakes in a short time? Based on the above P cycle pattern in the eiliang Bay, dredging which removes parts of sediments that accumulated FeP can be the fastest and the most effective method to cut off the pathway of FeP reductive dissolution. However, dredging would seriously destroy sediment ecological balance [55]. Algae fishing by mechanical equipment or by artificial means can directly remove algae cells from water during algae growth period [56], thus it has a certain advantage to mitigate the pathway of OP accumulation in suspended particles and FeP accumulation in sediments. Although algae fishing process is expensive and time-consuming, there are still many algae fishing activities being conducted with lots of manpower during algae bloom periods in China [56]. Nevertheless, chemical algae killing technology [57,58], flocculant method [59] can not

work well, because algae cells accompanied with P are just transferred temporarily from suspended particles to water or sediments, which would still go into the vicious P cycle later. From the perspective of breaking the vicious P cycle, economic, feasible and safe methods can be researched in the future.

## 5. Conclusions

This study found that concentration levels of TPP in water, OP in suspended particles, FeP in sediments and $PO_4^{3-}$ in pore water of Meiliang Bay, the cyanobacteria area, were basically higher than that in the middle of the lake and the east of the lake. The P cycle pattern of Meiliang Bay, which appeared overall high concentration levels of P forms in different mediums, were closely subjected to the influence of algae bloom and decomposition processes. The proportion IP and OP forms in suspended particles was totally contrary to that in sediments, indicating the occurrence of OP mineralization in sediments. The higher concentration and proportion of FeP in sediments of Meiliang Bay revealed the intensified effect of algae bloom and decomposition on sediment FeP accumulation. In pore water of Meiliang Bay, $PO_4^{3-}$ concentrations were positively correlated with $Fe^{2+}$ concentrations, and their diffusion fluxes from pore water to overlying water were significantly higher than that in the middle of the lake and the east of the lake. Consistent with these were the obvious high concentrations of DGT-$S^{2-}$ and the high relative abundances of FRBs and SRBs in sediments of Meiliang Bay. These indicated that algae bloom and decomposition processes also intensified sediment FeP reductive dissolution accompanying sediment P remobilization in Meiliang Bay, which were mainly driven by the coupled factors of anoxic sediment condition, sediment SRBs and FRBs activities. Thus, algae bloom event changes the natural P cycle in water environment system through intensifying the pathways of OP mineralization, FeP accumulation and FeP reductive dissolution in sediments. In addition, the high $Fe^{2+}$ concentration and the high Fe/P ratio in pore water of the east of the lake may promote to form a barrier of iron oxides in the P-O interface, preventing $PO_4^{3-}$ from spreading up overlying water. This study enhances the understanding of the vicious P cycle pattern in aquatic environment driven by algae bloom and decomposition, which should be considered when conducting eutrophication prevention and control measures on lakes.

**Supplementary Materials:** The following supporting information can be downloaded at: https://www.mdpi.com/article/10.3390/w14223607/s1, Table S1. Water quality parameters of Taihu Lake. Table S2. Molar Fe/P ratios and $PO_4^{3-}$, $Fe^{2+}$ diffusion fluxes from pore water to overlying water of Taihu Lake. Table S3. Abundance and diversity index of sediment bacteria communities of Taihu Lake. Figure S1. Concentrations of P forms in suspended particles of Taihu Lake. Figure S2. Heatmap plot of the top thirty bacterium at genus level in surface sediments of Taihu Lake. Figure S3. The Proportion of IP and OP in S-TP in suspended particles and sediments of Taihu Lake.

**Author Contributions:** Conceptualization, C.H.; investigation, Y.D. and N.S.; data curation, Y.D. and H.W.; formal analysis, Y.T. and T.D.; writing—original draft preparation, C.H.; writing—review and editing, C.H. and N.S.; project administration, C.H.; funding acquisition, C.H. All authors have read and agreed to the published version of the manuscript.

**Funding:** This research was funded by Natural Science Foundation of Jiangsu province in China [No. BK20190763] and the National Natural Science Foundation of China [No. 42007346]. And The APC was funded by Natural Science Foundation of Jiangsu province in China [No. BK20190763].

**Data Availability Statement:** Some or all of the data that support the findings of this study are available from the corresponding author upon reasonable request.

**Acknowledgments:** This research was supported by the Natural Science Foundation of Jiangsu province in China (No. BK20190763) and the National Natural Science Foundation of China (No. 42007346).

**Conflicts of Interest:** The authors declare no conflict of interest.

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
