# Peer review of "Algae Bloom and Decomposition Changes the Phosphorus Cycle Pattern in Taihu Lake"

_water, doi:10.3390/w14223607_

Round 1

Reviewer 1 Report

This presented manuscript discussed P cycle in aquatic environment of Taihu Lake under the influence of algae bloom events. It is an interesting and excellent work. This manuscript is well organized with worthwhile value for lake eutrophication treatment.

Some specific recommendations are listed as follows:

(1) What is DGT-S2-? Are they in pore water or sediments? they were confusedly described in Section 2.3 and Fig.4 and Fig.5

(2) Figure: the number of the figure is chaotic, such as Fig.1, Fig.S1, Fig.2, Fig.S2. If Fig.S1, Fig.2, Table S1 belong to supporting Information, put them in the supplementary files.

(3) 3.2 Section, the first paragraph: it is more appropriate to replace colons with single quotation marks in English writing

(4) Conclusion: the proportion IP and OP forms in suspended particles was totally contrary to that in sediments, indicating the occurrence of OP mineralization in sediments. —— please add figures about proportions of P forms in suspended particles and in sediments.

(5) section 2.1 and section 3.1: please clarify the sampling time. July 2020 or August 2020?

Author Response

(1) What is DGT-S2-? Are they in pore water or sediments? they were confusedly described in Section 2.3 and Fig.4 and Fig.5

Response: Thanks for your valuable comments.

DGT is an in situ technique for kinetic passive sampling measurement of analytes in sediments or wet soils over the deployment period. DGT-S2- refers to the labile sulfide in sediments adsorbed by AgI-DGT probe in 24 h.

We have supplied the interpretation of DGT-S2- in section 2.3.

We have revised the titles of Fig.4 and Fig.5 as show below.

 “Figure 4. Concentrations of PO43-, Fe2+ in pore water and DGT-S2- in sediments of Taihu Lake.”.

“Figure 5. Concentrations of PO43-, Fe2+ in the profile of pore water and DGT-S2- in the profile of sediment of Taihu Lake”.

(2) Figure: the number of the figure is chaotic, such as Fig.1, Fig.S1, Fig.2, Fig.S2. If Fig.S1, Fig.2, Table S1 belong to supporting Information, put them in the supplementary files.

Response:

We putted Fig.S1, Fig.S2, Table S1, Table S2, Table S3, in the supplementary file.

(3) 3.2 Section, the first paragraph: it is more appropriate to replace colons with single quotation marks in English writing

Response:

We have revised the sentence of “In water, TP concentrations were found in the following order: Meiliang bay (0.13~0.42 mg/L) > East of lake (0.18~0.28 mg/L) > Middle of lake (0.08~0.12 mg/L).” to “In water, TP concentrations were found in the order of ‘Meiliang bay (0.13~0.42 mg/L) > East of lake (0.18~0.28 mg/L) > Middle of lake (0.08~0.12 mg/L)’.”

(4) Conclusion: the proportion IP and OP forms in suspended particles was totally contrary to that in sediments, indicating the occurrence of OP mineralization in sediments. —— please add figures about proportions of P forms in suspended particles and in sediments.

Response:

We have supplied Fig.S3 entitled “The Proportion of IP and OP in S-TP in suspended particles and sediments of Taihu Lake” in the supplementary file.

Figure S3. The Proportion of IP and OP in S-TP in suspended particles and sediments of Taihu Lake

(5) section 2.1 and section 3.1: please clarify the sampling time. July 2020 or August 2020?

Response:

The sampling time was form July 31th to August 2th, 2020, so we wrote August 2020. We have revised it in Section 2.1.

Reviewer 2 Report

Overall comments:

The author compared P form characteristics and bacteria community structures in aquatic environment of the cyanobacteria area and non-cyanobacteria area of Taihu Lake, aiming to clear the new P cycle pattern disturbed by algae bloom and decomposition process. The conclusion showed that algae bloom event changed the natural P cycle in aquatic environment through intensifying the pathways of sediment OP mineralization, FeP accumulation and FeP reductive dissolution.

Generally, the topic of the manuscript is very interesting,it is well written and met the scope of this journal. I would like to recommend editor to accept it with minor revision.

Specific comments:

(1) Page 8, Line 256, Fig.5 showed the vertical distribution of PO43-, Fe2+ and DGT-S2- in the interface between overlying water and pore water, and inflection point for PO43-, Fe2+ were showed. However, there were little discussions about inflection points of PO43-, Fe2+ and DGT-S2- in the vertical direction.

(2) Some reference structures are incomplete. Please check if it is correct and, if necessary, change it throughout the whole manuscript.

(3) Page 8, Line 252, change ‘overling water’ to ‘overlying water’.

Author Response

(1) Page 8, Line 256, Fig.5 showed the vertical distribution of PO43-, Fe2+ and DGT-S2- in the interface between overlying water and pore water, and inflection point for PO43-, Fe2+ were showed. However, there were little discussions about inflection points of PO43-, Fe2+ and DGT-S2- in the vertical direction.

Response: Thanks for your valuable comments.

The synchronous vertical variation of PO43- and Fe2+ concentrations in pore water of three areas of Taihu Lake confirmed the occurrence of FeP reductive dissolution. It can be seen from Figure 5 that the most active areas of FeP reductive dissolution in sediments of Meiliang bay, Middle of lake, East of lake were concentrated at depths of -10 ~ -60 mm, -10 ~ -40 mm, -10 ~-40 mm. We have supplied these discusses in the section 4.2.

(2) Some reference structures are incomplete. Please check if it is correct and, if necessary, change it throughout the whole manuscript.

Response:

We have checked and revised the reference of the whole manuscript.

(3) Page 8, Line 252, change ‘overling water’ to ‘overlying water’.

Response:

We have revised this mistake.
